# Environmental and Anthropogenic Influences on Coliform Concentrations in the *Octopus insularis* Production Chain in the Veracruz Reef System, Gulf of Mexico

**DOI:** 10.3390/ani13193049

**Published:** 2023-09-28

**Authors:** Sarai Acuña-Ramírez, María de Lourdes Jiménez-Badillo, Gabriela Galindo-Cortes, Angel Marval-Rodríguez, María del Refugio Castañeda-Chávez, Christian Reyes-Velázquez, Hectorina Rodulfo-Carvajal, Marcos De Donato-Capote

**Affiliations:** 1Instituto de Ciencias Marinas y Pesquerías, Universidad Veracruzana, Hidalgo 617, Boca del Río 94290, Mexico; zs18015795@estudiantes.uv.mx (S.A.-R.); ggalindo06@gmail.com (G.G.-C.); avgelo7@gmail.com (A.M.-R.); 2Instituto Tecnológico de Boca del Río, Tecnológico de México, Carretera Veracruz-Córdova, km 12, Boca del Río 94290, Mexico; mariacastaneda@bdelrio.tecnm.mx (M.d.R.C.-C.); christianreyes@bdelrio.tecnm.mx (C.R.-V.); 3Escuela de Ingeniería y Ciencias, Tecnológico de Monterrey, Epigmenio González 500, Querétaro 76130, Mexico; hrodulfo2002@yahoo.es (H.R.-C.); marcosdedonato@yahoo.com (M.D.D.-C.)

**Keywords:** production chain, coliforms, octopus, total and fecal coliforms

## Abstract

**Simple Summary:**

The contamination of coastal waters, unhealthy conditions and inadequate handling practices tend to reduce the sanitary quality of fishery products, thus impacting its marketing. With this in mind, we investigate the microbiological quality of the *Octopus insularis* in each stage of the production chain, comprehending capture, post-capture, processing and commercialization, in terms of the presence of total and fecal coliforms at the Veracruz Reef System, Gulf of Mexico. The environmental and anthropogenic influence on the space–temporal concentration of coliforms were analyzed in sea water, fresh octopus, fresh water, ice and octopus, both packed in ice and boiled. Most relevant results indicated that coliforms are present in the octopus production chain, being highest in the marketing stage. The coliform concentration increased during the rainy season and was highest in the reefs closer to the coast, which has a major anthropogenic influence. These results point out the urgent need to implement an efficient cold chain with adequate handling practices to try to reverse these microbiological conditions and improve the octopus quality and food safety.

**Abstract:**

Coliforms are relatively common in aquatic environments, but their concentrations can be increased by environmental changes and anthropogenic activities, thus impacting fisheries resources. To determine the microbiological quality in the octopus production chain (capture, post-capture, processing and commercialization), total (TC) and fecal (FC) coliforms were quantified in sea water, fresh octopus, fresh water, ice and octopus in two presentations: packed in ice and boiled. Samples came from fishing zones Enmedio, Chopa and La Gallega at the Veracruz Reef System (VRS) during dry, rainy and windy seasons. The coliforms were determined using the most probable number technique (MPN). The most relevant results indicated that octopus packed in ice coming from the commercialization stage had FC levels >540 MPN/100 g, which exceeded the permissible limits (230 MPN/100 g). Therefore, these products present a risk for human consumption. Differences in FC were observed in octopuses between the three fishing zones (H = 8.697; *p* = 0.0129) and among the three climatic seasons, increasing during the rainy season, highlighting La Gallega with 203.33 ± 63 MPN (H = 7.200; *p* = 0.0273). The results provide evidence of the environmental and anthropogenic influences on coliform concentrations and the urgent need to implement an efficient cold chain throughout octopus production stages with adequate handling practices to reverse this situation.

## 1. Introduction

Foods of marine origin are important sources of protein of high biological value but are highly perishable if proper safety measures are not taken. The quality and safety of fishery products is essential for their commercialization [1,2]. The contamination of coastal waters caused by the continuous discharges of untreated industrial and domestic waste [3,4] and the sanitary and handling conditions of fishery resources are the principal causes of problems with food safety in these products. Monitoring at each stage of the production chain is recommended to prevent contamination and inhibit the multiplication of microorganisms that can cause gastrointestinal diseases in humans [5,6]. 

The octopus fishery in Mexico is important from an economic and social point of view, being the second most important fishery in the Gulf of Mexico and the Caribbean. [7,8]. The octopus fishery in the Veracruz Reef System (VRS) is an economic activity on which fishing communities depend [9], and it is also characterized by being related to reef systems, within a natural protected area [10]. The production chain of the octopus *Octopus insularis* in the VRS consists of four stages before consumption: capture, post-capture, processing and commercialization [11]. Artisanal fishing for octopus occurs in the VRS in the form of freediving with octopus hooks at depths of 0.5–5 m from March to July and September to December [10,12,13]. Post-capture handling consists of keeping the octopuses in fresh water until they are processed or commercialized. Boiling at high temperatures is the only process to which part of catches are subjected. Commercialization of the octopuses takes place on the beach shore, in local fish markets, restaurants and regional seafood markets in two presentations: boiled or packed in ice [14].

The commercialization of artisanal catches increases the risk of transferring alimentary infections due to limitations on the sanitary and handling conditions of fishery products, such as intermittent access to potable water, frequent breakdowns of freezing systems, and surface areas inadequate for processing raw materials [5,6,15]. There are protocols for the handling of fishery and aquaculture products from their capture in small boats through to commercialization [16], such as those used in Yucatán for the handling of *Octopus maya* [17]. Nevertheless, the handling practices and microbiological status in *O. insularis* production chain have not been evaluated, it being essential to know them to improve their quality and marketability.

Microbiological quality in fishery resources is determined by the presence and concentration of total and fecal coliforms. The habitat of both coliforms is the intestine [18], but these bacteria have the capacity to survive and multiply in other substrates, so high concentrations are indicative of poor sanitary practices [1]. In addition, it is possible that the concentration of FC may increase during the rainy season caused by the increment of discharges containing industrial and domestic wastes. The VRS is exposed to a high anthropogenic impact due to its proximity to a large urban area and one of the main ports of Mexico; therefore, an increase in fecal coliforms would be expected in the octopus production chain, thus affecting its safety.

Here, we present data on the microbiological quality of the *O. insularis* in each stage of the production chain and provide evidence of the environmental and anthropogenic influences on the space–temporal concentration of coliforms, sustaining the urgent need to implement an efficient cold chain with adequate handling practices to revert this situation.

## 2. Materials and Methods

### 2.1. Sampling Strategy

Two representative octopus fishing zones in the VRS (19°05′–19°14′ N and 95°93′–96°15′ W) were chosen for sampling: in the northern zone, considering the proximity to the coast and the port in La Gallega reef, and in the southern zone in Enmedio and Chopa reefs, considering the influence by anthropogenic activities and discharge from rivers (Figure 1). The influence by climatic seasons, namely dry (April to June), rainy (July to September) and the northern winds “windy” (October to March), were considered too. 

Samples of seawater, freshwater, octopuses, and ice from each stage of the production chain in the three-sample site and the three climatic seasons were taken. Capture stage included seawater samples from the water column at 150 cm depth, in duplicate, and stored in sterile containers of 250 mL capacity. They were kept refrigerated until processing. Also, three whole octopus freshly caught were preserved in sterile whirl-pak bags, on ice until processing. In post-capture stage, 100 mL freshwater, where the octopuses are stored before being sold, was sampled, and then preserved in sterile whirl-pak bags until its analyses. For the processing stage, three samples of the boiled octopus were taken, preserved in sterile conditions, and analyzed, while for the commercialization stage, three octopuses packed in ice and 250 g of the ice in which the octopuses are placed for exhibition and sale were sampled. All samples were collected, processed, and contrasted following the norms stipulated in APHA 2012 [19] and NOM-242-SSA1-2009 [20].

**Figure 1 animals-13-03049-f001:**
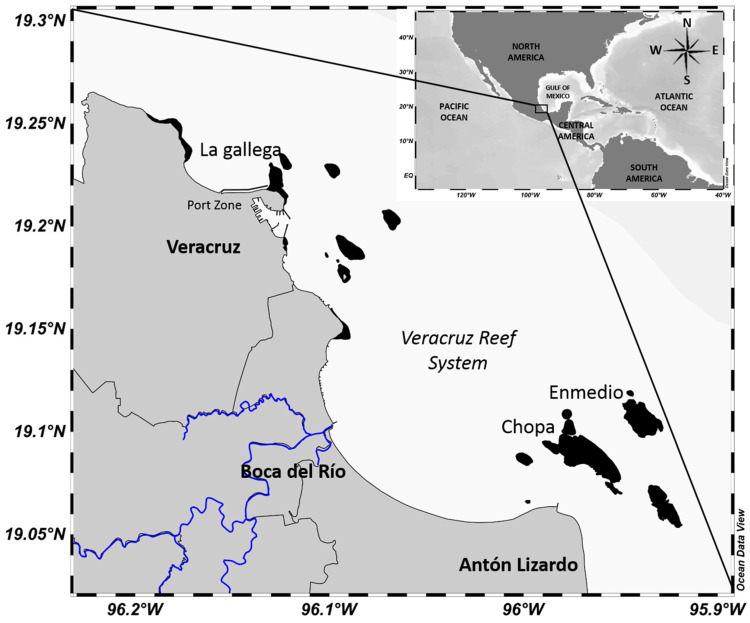
Study area. Veracruz Reef System (VRS). Octopus fishing zones: La Gallega reef (northern), Chopa and Enmedio Reefs (southern). Jamapa river in blue lines. Map prepared using Ocean Data View software (Version 4.7.8) [21].

### 2.2. Determination of Fecal and Total Coliforms in Sea Water, Fresh Water, Ice and Octopuses

For the water samples (sea, fresh and ice) three serial dilutions were made by transferring 1 mL of the sample in 9 mL of saline peptonized water to obtain 3 concentrations (10^−1^, 10^−2^ and 10^−3^). In the case of octopus, 10 g of muscle was taken and homogenized in 90 mL of saline peptonized water, from which serial dilutions were performed for presumptive and confirmatory tests, in the same way as was described for water samples.

The microbiological analysis of seawater, freshwater, ice and octopuses were carried out in accordance with the protocols described by the U.S. Food and Drug Administration [22], which specify three serial dilutions per sample. These dilutions were used to perform presumptive and confirmative tests for total (TC) and fecal (FC) coliforms, respectively. The enumeration of TC and FC was realized via the Most Probable Number (MPN) technique, according to APHA [23], using the following media: Lauryl Sodium Sulfate Broth (presumptive test for the total coliforms), Lactose Bile 2% Brilliant Green Broth in 37 °C (confirmative test for the TC and the presumptive for FC) and *Escherichia coli* Broth in 44 °C (confirmation for the FC).

From the number of positive tubes in the presumptive and confirmatory tests, the MPN of TC and FC in 100 mL of sample was calculated, referring to the MPN statistical tables. In the case that the presumptive test did not show turbidity and gas production, the minimum value expressed in the table was reported.

The MPN for TC and FC in each sample was obtained in accordance with norm NOM-112-SSA1-1994 [24] and then contrasted to the values established in norm NOM-201-SSA1-2015 [25], which refers to the quality of water and ice for human consumption, and norm NOM-242-SSA1-2009 [20], which specifies the quality standards for fresh, refrigerated, frozen, and processed marine products.

### 2.3. Statistical Analysis

Normality and homoscedasticity tests were applied on TC and FC data to discern about the use of parametric or non-parametric tests. Results led us to choose the Kruskal–Wallis test to determine the influence of the climatic seasons and fishing zones on the MPN for TC and FC on each stages of the production chain. In the cases where significant differences were found, multiple contrast tests were applied following the Dunn test. All tests were carried out considering a significance level of α = 0.05. Analyses were performed with the STATA package version 13.1.

### 2.4. Environmental Parameters

To analyze climatic variation, we utilized the database MOHICOVER (Monitoring of Hydrography and Currents in Veracruz), which contains monthly data of temperature and salinity of the study area (VRS) during 2019. Also, we collected precipitation data and water discharge flow from the Jamapa River based on official CONAGUA statistics (Comisión Nacional del Agua, National Water Commission) from January 2019 to December 2020.

## 3. Results

### 3.1. Total Coliforms by Stage of Octopus Production Chain

The spatial and temporal analysis of the TC and FC show that the seawater did not detect statistically significant differences in TC concentrations between the three climatic seasons in the reefs evaluated (Table 1). The highest MPN value for TC in the fresh octopus was obtained in La Gallega reef in the capture stage during the rainy season (Figure 2).

The MPN for TC in the fresh octopuses captured in the different climatic seasons presented statistically significant differences within the reefs sampled: Enmedio (H = 7.20; *p* < 0.05), Chopa (H = 8.526; *p* < 0.05) and La Gallega (H = 7.200; *p* < 0.05). In contrast, there were no significant differences in the MPN for TC among the reefs (H = 1.734; *p* > 0.05) (Table 1).

With respect to the post-capture stage, statistically significant differences (H = 7.200 *p* < 0.05) were observed in the MPN for TC isolated in fresh water in the different climatic seasons, with higher values in the dry season, while for the processing stage, no statistically significant differences (H = 1.046; *p* > 0.05) were observed between the MPN for TC of the boiled vs. fresh octopuses (Figure 2).

Regarding the commercialization stage, the MPN for TC obtained from the octopuses packed in ice was higher than in the boiled octopuses, with statistically significant differences between the two sets of samples (H = 8.750; *p* <0.05). For the samples of ice, the same high MPN value for TC was found during the three climatic seasons (Figure 2).

### 3.2. Fecal Coliforms by Stage of Octopus Production Chain

The spatial and temporal FC concentrations in sea water from the capture stage did not reveal statistically significant differences (H = 0.762; *p* > 0.05) in the reefs evaluated (Table 2). The MPN of CF in fresh octopus captured in the different climatic seasons presented statistically significant differences both between (H = 8.697; *p* < 0.05) and within reefs sampled: Enmedio (H = 3.857; *p* < 0.05), Chopa (H = 5.44; *p* ≤ 0.05) and La Gallega (H = 7.200; *p* < 0.05) (Figure 3).

The post-capture stage analyses identified statistically significant differences in the MPN for FC isolated in fresh water in the different climatic seasons (H = 7.200; *p* < 0.05), with a marked increase in this value for the fresh water in the rainy season, while for the processing stage, no statistically significant differences between the MPN for FC in the boiled vs. fresh octopuses (H = 0.198; *p* > 0.05) (Figure 3). 

Regarding the commercialization stage, the MPN for FC obtained from the octopuses packed in ice was higher than in the boiled octopuses, with statistically significant differences between these samples (H = 8.2611; *p* < 0.05). Moreover, the ice used to preserve the octopuses presented FC values (>800 MPN/100 g) above allowable levels (230 MPN/100 g) in all three climatic seasons (Table 2). 

### 3.3. Environmental Parameters

The analysis of temperature anomalies in the sea indicated below-average values from February to May and from November to December 2019, but higher values from June to October. The maximum (29.9 °C) and minimum (21.8 °C) values for this variable were obtained in August and March, respectively. In relation to salinity, variations from average values were observed throughout the year. The lowest value was registered in April 2019 (30.8 UPS), the highest in October 2019 (35.5 UPS). Precipitation in 2020 was higher than in 2019, with above-average increases in October 2019 and September 2020, and periods of low precipitation between December and April in both years. A similar tendency was found for the rate of discharge from the Jamapa River, where we found an increase from June/July to October in both 2019 and 2020 (Figure 4).

## 4. Discussion

The analysis of coliform concentrations in the different stages of the octopus production chain in the VRS revealed an increased tendency in the MPN for both total (TC) and fecal (FC) coliforms in the octopuses from capture to the commercialization stage. In most of the cases, both values exceeded the allowable levels stipulated in the norm NOM-242-SSA1-2009 [20]. These results provide evidence of the failure in the good handling practices on these fishery resource. In addition, the direct discharge of untreated wastewater into seawater can be a source of fecal contamination to which octopuses are exposed.

The highest MPN of FC values in all samples of fresh octopus captured from La Gallega, during the rainy season, could be due to the greater anthropogenic impact, such as direct discharges of untreated wastewater [26], intense maritime traffic [27], pollution from oil spills, industrial discharges, domestic effluents, etc. [28,29,30], that these reefs receive owing to their proximity to a metropolitan area and port zone. This increase in coliforms during the rainy season has been evidenced in several studies, where it is associated with an increase in runoff and wastewater towards the coasts [31]. In addition, this increase in discharges decreases the salinity and increases the temperature of the system, thus favoring bacterial growth [32,33].

A previous study had warned that La Gallega reef is one of the most susceptible to suffering a high degree of degradation of its waters, which can cause diseases in fish and coral [34]. The results reported herein provide evidence of the impact of these conditions on the octopuses extracted in this zone. 

In the post-capture stage, the MPN values for TC and FC in the fresh water that fishers use to preserve their catches prior to commercialization were also found to exceed the allowable limits for potable water (0 MPN for TC) [35]; also, these levels were highest in the rainy season, when according to Cho et al. [36], ambient temperatures tend to rise. The inadequate post-capture handling practices, the limited availability of potable water in the zone and the use of unsuitable recipients provoked high TC and FC concentrations.

The handling practices of the octopus resource in both zones of the VRS do not comply with the norms NOM-242-SSA1-2009 [35], OIRSA [16] and SAGARPA-SENASICA [37]. We detected a breaking off in the cold chain from capture to commercialization stages. In addition, the CF values obtained in the ice suggest that it does not come from potable water. The exhibition of catches for commercialization in trays with no incline and with water-ice that does not comply with norm NOM 251-SSA1-2009 [20] can be another source of contamination of the octopuses. Even in boiled octopus, our analyses found values for TC and FC that exceeded allowable levels, therefore representing a risk for human health when consumed, which, according to Steffen et al. [38], can cause gastrointestinal infections. While the study did not apply a confirmatory test for the presence of *Escherichia coli* and, thus, the MPN could pertain to this species or some other in the group of thermotolerant bacteria, the presence of fecal coliforms is an indicator of anthropogenic contamination that must be addressed.

Despite having a good cooking process, there is “cross contamination” due to the lack of instruments and materials used exclusively for handling cooked products, sharing scales, trays, knives, boards and ice with other uncooked seafood products. In the same way, the high CF values in the ice during all the months sampled indicate that they do not come from potable water; in addition, the ice was used only in the commercialization stage, in non-inclined trays, which means that as the ice melts, water ice forms and is not at the temperature determined for the maintenance of the resources and the inhibition of bacterial growth. It is necessary to use and continually replace ice throughout the octopus production chain to reduce the growth of fecal coliforms.

In addition to anthropogenic factors, which influence the number of total and fecal coliforms, the VRS presents a stratification of the water column that differs between seasons (rainy, dry, windy) [39]. Temperature is a determining factor in bacterial growth, having a directly proportional relationship. In warm climates, bacterial growth is faster, observing that during the months of July to September, the highest temperatures are reported, relating this value to the highest concentration peaks of fecal coliforms found in this study. Higher temperatures are also recorded during the rainy season with increased water flow discharges, which increases the concentration of nutrients and wastewater discharges that bring with them fecal coliforms.

The results of this study provide clear evidence that in order to correct this situation, the following issues must be addressed: treating all residual waters before discharging them into the sea; maintaining the cold chain throughout all stages of the octopus production chain; improving fishery infrastructure and the access to potable water; implementation of good handling and monitoring practices; and providing technical orientation and support to all the personnel involved in the octopus production chain. These measures should be complemented by the practices in Shawyer and Medina [40], DOF [26] and norm NOM-242-SSA1-2009 [35]. Practical measures include daily cleaning of boats, use of coolers with lids for the octopus caught, ice from potable water, personnel hygiene and the use of equipment exclusively for cooked products. Finally, increasing the awareness of all the personnel involved of the risks of contamination by biological, physical or chemical factors [2] will contribute to improving the quality and food safety of this product.

## 5. Conclusions

Total and fecal coliforms are present in the octopus production chain from the VRS, being highest in the marketing stage, with FC values above the permissible values even in cooked octopus. The most affected octopuses were those captured in coastal waters near reefs with the greatest anthropogenic pressure. During the rainy season, the concentration of FC increases. It is necessary to strengthen the cool chain and implement good handling practices throughout the octopus production chain to improve the quality and food safety of this product.

## Figures and Tables

**Figure 2 animals-13-03049-f002:**
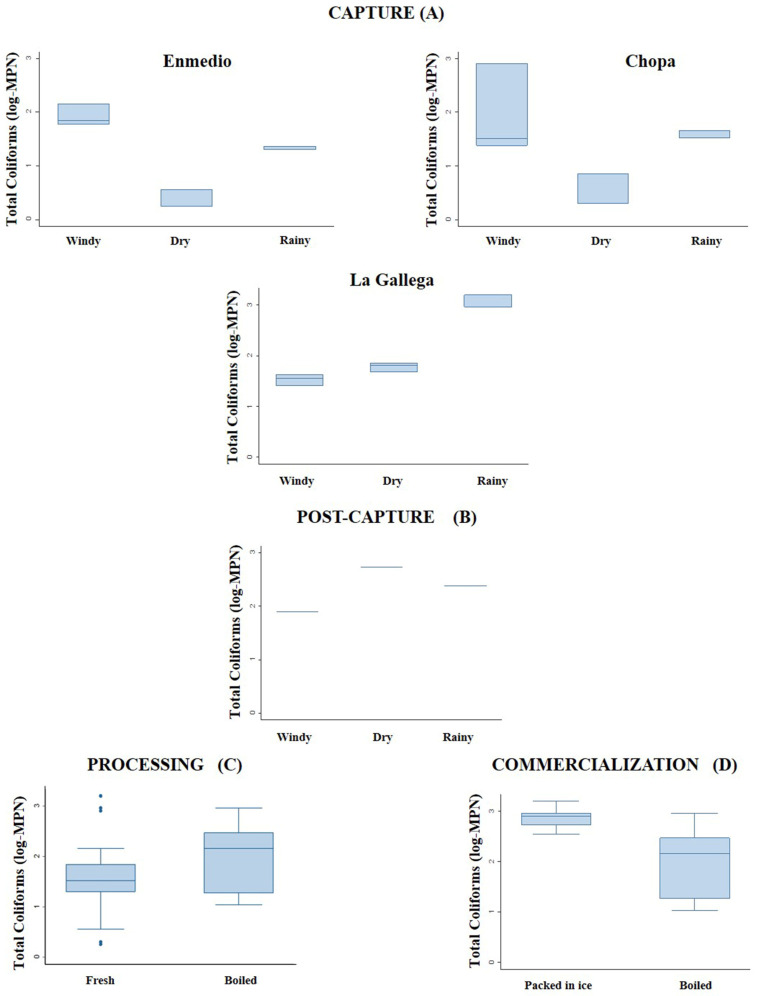
Variation in the most probable number (MPN) of total coliforms (TC) isolated from: (**A**) fresh octopus capture in the Enmedio, Chopa and La Gallega reefs, (**B**) freshwater in post-capture stage in different climatic seasons, (**C**) boiled and fresh octopus in processing stage and (**D**) packed-in-ice and boiled octopus from commercialization stage.

**Figure 3 animals-13-03049-f003:**
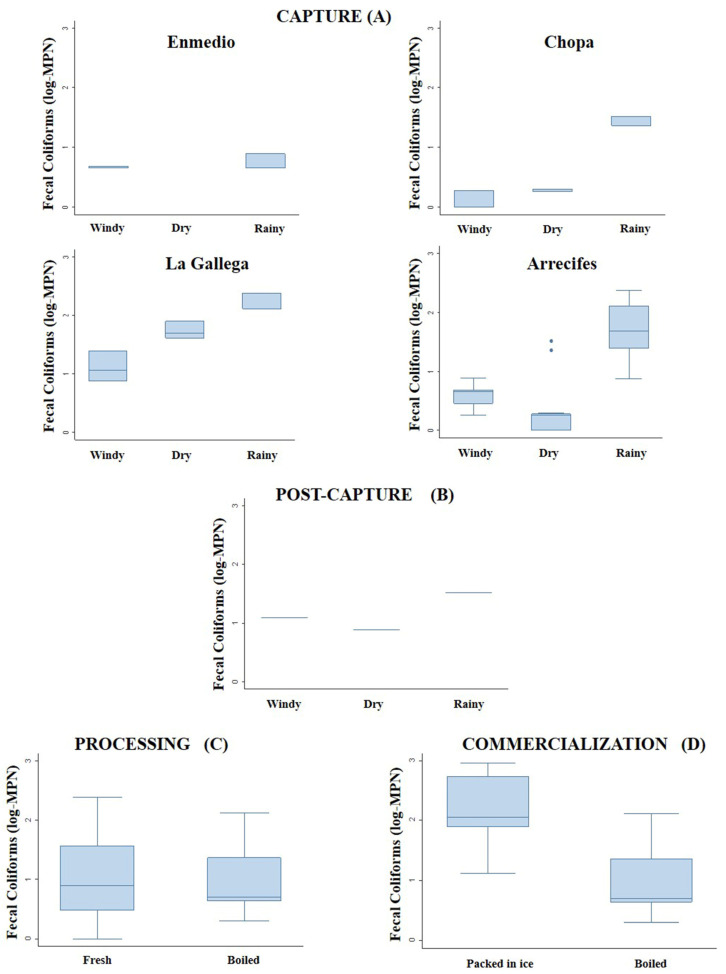
Variation in the most probable number (MPN) of fecal coliforms (FC) isolated from: (**A**) fresh octopus capture in the Enmedio, Chopa and La Gallega reefs, (**B**) freshwater in post-capture stage in different climatic seasons, (**C**) boiled and fresh octopus in processing stage and (**D**) packed-in-ice and boiled octopus from commercialization stage.

**Figure 4 animals-13-03049-f004:**
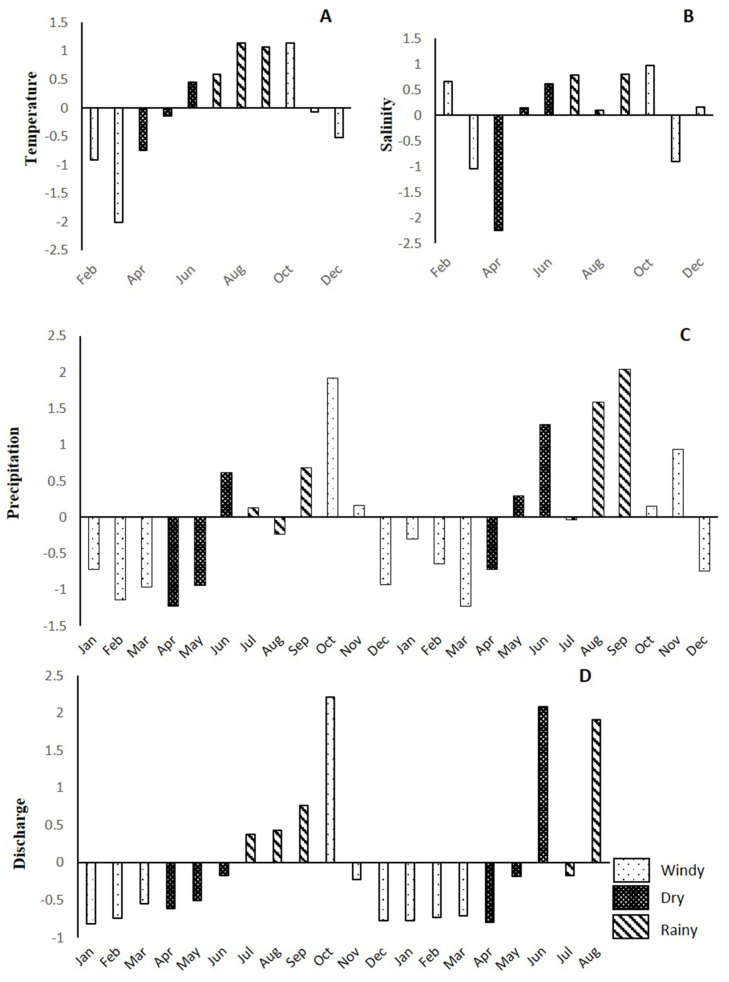
Anomalies of temperature (**A**) and salinity (**B**) in the Veracruz Reef System during 2019. Monthly precipitation anomalies (**C**) in Veracruz from January 2019 to December 2020. Jamapa River discharge rate (**D**) expressed in anomalies from January 2019 to August 2020.

**Table 1 animals-13-03049-t001:** Results of the Kruskal–Wallis tests, examining the mean differences in the concentration of total coliforms (TC MPN ± σ) isolated in the samples of seawater, freshwater, fresh octopus, octopus boiled, octopus packed in ice, and ice with regard to the space–temporal distribution and the stages of the octopus production chain (capture, post-capture, processing and commercialization). Identical superscript letters mean no significant differences (NS). Asterisks indicate statistically significant differences (*p* < 0.05).

Stages	Zone	Climatic Seasons	Sample	TC (MPN ± σ)	N	*p*
Capture	Southern zone (Enmedio)	Dry	Sea water	14.4 ± 13.9 ^a^	6	0.2305 NS
		Rainy	Sea water	1.8 ± 0 ^a^	6	
		Windy	Sea water	8.15 ± 8.19 ^a^	12	
		Dry	Fresh octopus	2.4 ± 1.04 ^a^	3	0.0273 *
		Rainy	Fresh octopus	22 ± 1.73 ^b^	3	
		Windy	Fresh octopus	91.1 ± 45.92 ^c^	6	
	Southern zone (Chopa)	Dry	Sea water	3.15 ± 1.48 ^a^	6	0.0608 NS
		Rainy	Sea water	1.8 ± 0 ^a^	6	
		Windy	Sea water	25.15 ± 25.47 ^a^	12	
		Dry	Fresh octopus	3.73 ± 3.00 ^a^	3	0.0141 *
		Rainy	Fresh octopus	37.33 ± 7.51 ^b^	3	
		Windy	Fresh octopus	286.83 ± 447.55 ^c^	3	
	Northern zone (La Gallega)	Dry	Sea water	1.8 ± 0 ^a^	6	1 NS
		Rainy	Sea water	1.8 ± 0 ^a^	6	
		Windy	Sea water	1.8 ± 0 ^a^	6	
		Dry	Fresh octopus	61.33 ± 12.22 ^a^	3	0.0273 *
		Rainy	Fresh octopus	1373.33 ± 392.59 ^b^	3	
		Windy	Fresh octopus	34.73 ±8.81 ^c^	3	
	Between zones (Southern, Northern)		Fresh octopus			0.4202 NS
Post-capture	Southern zone	Dry	Freshwater	540 ± 0 ^a^	3	0.0273 *
		Rainy	Freshwater	240 ± 0 ^b^	3	
		Windy	Freshwater	79 ± 0 ^c^	3	
	Northern zone	Not applicable	Not applicable			
Processing	Both zones	Not applicable	Boiled octopus	226.06 ± 290.82 ^a^	9	0.3063 NS
			Fresh octopus	212.53 ± 457.61 ^a^	36	
Commercialization	Both zones	Not applicable	Boiled octopus	226.06 ± 290.82 ^a^	9	0.003 *
			Octopus packed in ice	853.93 ± 463.87 ^b^	9	
			Ice	1600 ± 0	9	

**Table 2 animals-13-03049-t002:** Results of the Kruskal–Wallis tests, examining the mean differences in the concentration of fecal coliforms (FC MPN ± σ) isolated in the samples of seawater, freshwater, fresh octopus, octopus boiled, octopus packed in ice, and ice with regard to the space–temporal distribution and the stages of the octopus production chain (capture, post-capture, processing and commercialization). Identical superscript letters mean no significant difference (NS). Asterisks indicate statistically significant differences (*p* < 0.05).

Stages	Zone	Climatic Seasons	Sample	FC (MPN ± σ)	N	*p*
Capture	Southern zone (Enmedio)	Dry	Sea water	4.5 ± 0 ^a^	3	0.3827 NS
		Rainy	Sea water	0 ^a^	3	
		Windy	Sea water	1.8 ± 0 ^a^	3	
		Dry	Fresh octopus	0 ^a^	3	0.0495 *
		Rainy	Fresh octopus	6.7 ± 1.91 ^b^	3	
		Windy	Fresh octopus	4.7 ± 0 ^c^	3	
	Southern zone (Chopa)	Dry	Sea water	0	6	Not applicable
		Rainy	Sea water	0	6	
		Windy	Sea water	0	6	
		Dry	Fresh octopus	1.87 ± 0.12 ^a^	3	0.0201 *
		Rainy	Fresh octopus	29.67 ±5.77 ^b^	3	
		Windy	Fresh octopus	1.3 ± 0.52 ^c^	3	
	Northern zone (La Gallega)	Dry	Sea water	0	6	Not applicable
		Rainy	Sea water	0	6	
		Windy	Sea water	0	6	
		Dry	Fresh octopus	56.33 ± 20.03 ^a^	3	0.0273 *
		Rainy	Fresh octopus	203.33 ± 63 ^b^	3	
		Windy	Fresh octopus	14.5 ± 8.89 ^c^	6	
	Between zones (Southern, Northern)		Fresh octopus			0.0129 *
Post-capture	Southern zone	Dry	Freshwater	7.8 ± 0 ^a^	3	0.0273 *
		Rainy	Freshwater	33 ± 0 ^b^	3	
		Windy	Freshwater	12.4 ± 0 ^c^	3	
	Northern zone	Not applicable	Not applicable			
Processing	Both zones	Not applicable	Boiled octopus	18.91 ± 36.27 ^a^	12	0.6566 NS
			Fresh octopus	21.50 ± 46.65 ^a^	29	
Commercialization	Both zones	Not applicable	Boiled octopus	18.91 ± 36.27 ^a^	12	
			Octopus packed in ice	266.72 ±339.76 ^b^	12	0.0041 *
			Ice	818.75 ± 816.03 ^c^	12	

## Data Availability

Data supporting the conclusions of this study are available from the corresponding author, [J.-B.M.d.L]. The data are not publicly available because they contain data that have not yet been published. Furthermore, it could compromise the privacy of the fishermen who collaborated in the research.

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
