# Peer review of "Environmental and Anthropogenic Influences on Coliform Concentrations in the Octopus insularis Production Chain in the Veracruz Reef System, Gulf of Mexico"

_animals, 2023, doi:10.3390/ani13193049_

Round 1
Reviewer 1 Report
This manuscript presented data on the environmental and anthropogenic influences on coliform concentrations in the octopus Octopus insularis production chain in the Veracruz Reef System, Gulf of Mexico. The manuscript is easy to read and understand, however, there are some several scientific concerns about this manuscript.
1. In section “2.1. Sampling strategy”, the specific months of the three climates including dry, rainy and windy, should be indicated.
2. In section “2.4. Environmental parameters”, the monthly data of temperature and salinity of Veracruz Reef System were only collected during 2019, is the data representative? And a similar question to the data of the precipitation and water discharge flow.
3. A significantly increase of the precipitation in November 2020 was presented in Figure 2, Line 147, the statement “periods of low precipitation between November and April in both years” is not correct.
4. Table 1, the sea water did not detect statistically significant differences in TC concentrations between the dry (41.4±43.39) and rainy (1.8±0) in the Enmedio, please check the accuracy of the data again.
5. Line 152, there are some problems in Figure 2. The Figures could be subdivided as A, B, C, D, and the legend also accordingly. The horizontal coordinate of Figure 2 is obscured and the months of different years should be clearly distinguished.
6. In Figure 3(A), the ordinate is obscured.
7. The boiling process can kill a certain amount of microorganisms. In Figure 3(C), would you please tell me why the total coliforms isolated from fresh octopus was higher than that the boiled one.
From the above considerations, there is the need to produce something more compelling than what the manuscript currently contains. The author should provide a reasonable explanation, otherwise it will not meet the requirements of the journal.
Minor editing of English language required.
Author Response
We would like to thank the comments and suggestions made by the referees, with which we fully agree and whose inclusion in this new version we consider contributed substantially to improve its quality.
This version of the manuscript includes attention to all comments made by the referees according with the following details:
Reviewer 1
- In section "2.1. Sampling strategy", the specific months of the three climates including dry, rainy and windy, should be indicated
The specific months of the climatic seasons were included. You can check it out in line 94 to 96
- In section "2.4. Environmental parameters" the monthlv data of temperature and salinitv of Veracruz Reef System were only collected during 2019, Is the data representative? And a similar question to the data of the precipitation and water discharage flow
In fact, the monthly data of temperature and salinity of Veracruz Reef System were only collected during 2019 due to the COVID 19 pandemic that prevent its collection during 2020. While precipitation and water discharge data were obtained despite the pandemic during both years, since they did not require field work from us but consultation through official sources.
- A significantly increase of the precipitation in November 2020 was presented in Figure 2, Line 147, the statement “periods of low precipitation between November and April in both years” is not correct.
Thanks for the observation, we made a mistake. The text was changed “low precipitation between December and April in both years”. You can check this on the lines 233 to 235.
- Table 1, the sea water did not detect statistically significant differences in TC concentrations between the dry (41.4±43.39) and rainy (1.8±0) in the Enmedio, please check the accuracy of the data again.
Thanks for the observation. We noted that there was a typographical error in the original base, so we proceeded to perform the analysis again, obtaining a new data for coliform values in seawater at Enmedio reef in Table 1, without modifying the significance of the data.
- Line 152, there are some problems in Figure 2. The Figures could be subdivided as A, B, C, D, and the legend also accordingly. The horizontal coordinate of Figure 2 is obscured and the months of different years should be clearly distinguished.
Thank you for your correction. The figure 2 in the original version, now figure 4 in the new version was subdivided into A, B, C and D and, also, was modified so that the names of the months were not covered by the bars.
- In Figure 3(A), the ordinate is obscured.
The figure 3(A) of the original manuscript (figure 2 of the new version) was arranged to clarify the ordinate. Line 175
- The boiling process can kill a certain amount of microorganisms. In Figure 3(C), would you please tell me why the total coliforms isolated from fresh octopus was higher than that the boiled one.
Total coliforms isolated from boiled octopus was higher than fresh octopus due to cross contamination caused by lack of the personal hygiene and the use of the same tools (knife, table, scale, bags, etc.) to handle fish and mollusk fresh and cooked.
Reviewer 2 Report
animals-2574098
“Environmental and anthropogenic influences on coliform concentrations in the octopus Octopus insularis production chain in the Veracruz Reef System, Gulf of Mexico”
In this manuscript, the microbiological quality of the Octopus insularis in each stage of the production chain, comprehending capture, post-capture, processing, and commercialization, were investigated in terms of presence of total and fecal coliforms. Evaluations were carried out by most probable number method in sea water, fresh octopus, fresh water, ice and octopus, both packed on ice and boiled. These samples were collected at the Veracruz Reef System, Gulf of Mexico. The environmental and anthropogenic influences on the space-temporal concentration of coliforms were highlighted, pointing out the urgent need to implement an efficient cold chain with adequate handling practices, try to revert worring microbiological conditions.
The methods are well described and can allow for a possible repetition of the experiments. Some more reference of international resonance could add information useful for the whole study.
The manuscript describes an interesting approach that highlights an important aspect for human health. This study is important and the results obtained can be referred to other realities in other countries. Thus, it has evidenced that the observations of rules of hygiene and of cold chain can help in maintaining safety of food.
Environmental parameters were the last in the M&M section and the first in the Results section, the order should be respected.
Again, the environmental parameters were poorly or not discussed in the Discussion section. Relationships between environment and the possible development of total and fecal coliforms could represent a significant aspect adding information, in particlular considering increasing temperatures due to climate change.
Revisions
Line 89: ‘was sample’ change to ‘was sampled’;
Line 112:’ test for the TC and the presumptive for FC’ maybe it could be useful to add the different temperatures, 37°C and 44°C for growth of these two bacterial groups;
Line 112: ‘Escherichia coli’ change to Italics;
Lines 147-149: ‘A similar 147 tendency was found for the rate of discharge from the Jamapa River, where we found an 148 increase from June/July to October in both 2019 and 2020 (Figure 2).’ The environmental parameters of Jamapa River have been reported in Figure 2, but no further discussion was provided at this regard.
Figure 3 and Figure 4 both present the y axis with ‘NMP’, please check;
Line 271: ‘Is necessary to …’ change to ‘It is necessary to …’.
The English form of the manuscript needs a minor revision.
Author Response
We would like to thank the comments and suggestions made by the referees, with which we fully agree and whose inclusion in this new version we consider contributed substantially to improve its quality.
This version of the manuscript includes attention to all comments made by the referees according with the following details:
Reviewer 2
We are agree with all your suggestions. Thank you.
Environmental parameters were the last in the M&M section and the first in the Results section, the order should be respected
In results, the section 3.1 Environmental parameters was moved to the bottom of the results to be consistent with the order in which the sections are presented in material and methods section. Because of this, the following changes have been made
|
Original manuscript
|
New manuscript version |
|
3.1. Environmental parameters |
3.1. Total coliforms by stage of octopus production chain |
|
3.2. Total coliforms by stage of octopus production chain |
3.2. Fecal coliforms by stage of octopus production chain |
|
3.3. Fecal coliforms by stage of octopus production chain |
3.3. Environmental parameters |
|
Figure 3 |
Figure 2 |
|
Figure 4 |
Figure 3 |
|
Figure 2 |
Figure 4 |
The environmental parameters were poorly or not discussed in the Discussion section. Relationships between environment and the posible development of total and fecal coliforms could represent a significant aspecto adding information, in particular considering increasing temperaturas due to climate change
The environmental parameters were more widely discussed considering the relationship between the environment and the possible development of the coliforms.
Line 89: “was sample” change to “was sampled”
The change indicated in line 89 “was sampled” was made, this appears in the new version on the line 103
Line 112 “test for the TC and the presumptive for FC” maybe it could be useful to add the different temperstures, 37oC and 44oC for growth of these two bacterial groups
Line 112 “Escherichia coli” change to italics
The temperatures 37 oC y 44 oC indicated in line 112, were included. Also Escherichia coli was changed to italics. You can check it out in lines 127 to 129 of the new version of the manuscript.
Lines 147-149
The text in lines 147-149 was adjusted and the discussion on environmental parameters was improved. You can check it out in lines 233 to 235 of the new version of the manuscript. The discussion about environmental parameters was improved. You can check it in lines 293 to 301 of the new version of the manuscript.
Figure 3 and figure 4
Thanks by the observation. The y axis of the figures 3 and 4 in the original version with regard to NMP, were changed to MPN. See figures 2 and 3 of the new version.
Line 271
The text in line 271 Is necessary to, was changed to It is necessary to. You can check it out in line 320 of the new version.
Reviewer 3 Report
Dear authors,
Overall the article is quantitatively consistent and supported by an appropriate methodology.
However, you could improve the quality of the article by adding further data and explaining in depth following aspects: a. What are the sources of faecal contamination of the seawater in the area where the specimens of O. insularis are collected? b. How do the authors explain the presence of a large number of coliforms after capture? Did the authors detect the source(s) of contamination in this case? what are practical measures to reduce contamination and ensure the safety of comsumption of octopuses? c. What is the reason that the ice contains coliforms above the allowed limit (lines 215-217)?
With best regards
English is fine.
Author Response
We would like to thank the comments and suggestions made by the referees, with which we fully agree and whose inclusion in this new version we consider contributed substantially to improve its quality.
This version of the manuscript includes attention to all comments made by the referees according with the following details:
Reviewer 3
We appreciate your questions, as they allowed us to conduct and complement the discussion section. We hope we have clarified all the points.
1. What are the sources of fecal contamination of the seawater in the area where the specimens of O. insularis are collected?
We think that the direct discharge of untreated sewage into seawater from domestic, hotel and industrial waste may be a source of fecal contamination to which octopuses are exposed. Line 250 to 251 of the new version of the manuscript.
2. How do the authors explain the presence of a large number of coliforms after capture?. Did the authors detect the sources of contamination in this case? What are practical measures to reduce contamination and ensure the safety of comsumption of octopuses?
A common practice in the study area is to place octopuses after capture in fresh water (usually non-potable). After that, the product is handled without hygienic measures, and the same utensils are used to handle cooked and raw organisms. Also, we detect cross-contamination at the commercialization stage where the same scale and trays were used for mollusks, crustaceans, and fresh fish. Lines 265 to 292.
In view of this situation, we suggest treating all residual waters before discharging them into the sea; maintaining the cold chain throughout all stages of the octopus production chain; improving fishery infrastructure and the access to the potable water, implementation of good handling and monitoring practices; and providing technical orientation and support to all the personnel involved in the octopus production chain. Other practical measures include daily cleaning of boats, use of coolers with lids for the octopus caught, ice from potable water, personnel hygiene and the use of equipment exclusively for cooked products. Lines 302 to 313.
3. What is the reason that the ice contains coliforms above the allowed limit?
We believe that the availability of non-potable water in the area, with which the ice is made, is the cause of the appearance of coliforms in the ice, also the ice was in trays not inclined, which did not discard the water that was forming as a result of the thawing of the ice, forming water ice not reaching the optimum temperatures to maintain the resources and inhibit bacterial growth. Line 286 to 291
In addition to this, the figure 1 was changed. The new map was developed with Ocean Data view software (Schilitzer, 2023). This reference was included in the corresponding section.
Round 2
Reviewer 1 Report
Through the modification, the article has been improved greatly. This article is recommended for acceptance.